# Characterisation and Modelling of an Artificial Lens Capsule Mimicking Accommodation of Human Eyes

**DOI:** 10.3390/polym13223916

**Published:** 2021-11-12

**Authors:** Huidong Wei, James S. Wolffsohn, Otavio Gomes de Oliveira, Leon N. Davies

**Affiliations:** 1College of Health and Life Sciences, Aston University, Birmingham B4 7ET, UK; l.n.davies@aston.ac.uk; 2Rayner Intraocular Lenses Limited, Worthing BN14 8AQ, UK; otaviogomes@rayner.com

**Keywords:** silicone rubber, biomechanical, hyper-elastic, constitutive model, FEA

## Abstract

A synthetic material of silicone rubber was used to construct an artificial lens capsule (ALC) in order to replicate the biomechanical behaviour of human lens capsule. The silicone rubber was characterised by monotonic and cyclic mechanical tests to reveal its hyper-elastic behaviour under uniaxial tension and simple shear as well as the rate independence. A hyper-elastic constitutive model was calibrated by the testing data and incorporated into finite element analysis (FEA). An experimental setup to simulate eye focusing (accommodation) of ALC was performed to validate the FEA model by evaluating the shape change and reaction force. The characterisation and modelling approach provided an insight into the intrinsic behaviour of materials, addressing the inflating pressure and effective stretch of ALC under the focusing process. The proposed methodology offers a virtual testing environment mimicking human capsules for the variability of dimension and stiffness, which will facilitate the verification of new ophthalmic prototype such as accommodating intraocular lenses (AIOLs).

## 1. Introduction

The eye’s crystalline lens capsule is a membrane with a thickness ranging from 5 to 30 µm [1,2,3], forming the capsular bag, which encompasses the lens substance. A primary biomechanical function of the in vivo lens capsule is to facilitate the mechanism of ocular accommodation [4]. According to Helmholtz [5], the ciliary muscle contracts when the eye focuses on a near target, relaxing the zonules attached to the lens equator, and enabling the lens surfaces to become more prolate (curved) for increased optical power (accommodation). Conversely, relaxation of the ciliary muscle, to view distant objects, causes centrifugal tensioning of the zonules and a corresponding flattening the lens, leading to decreased optical power (dis-accommodation). During the accommodating process, given its high Young’s Modulus (of 1 MPa) compared to the lens substance (of 1 Pa), the capsular bag is able to mould the shape of the internal lens [6,7]. It has been found that the biomechanical properties of lens capsule are relatively less affected by age, except the geometric inhomogeneity of the thickness distribution [3,8].

The subjective visual difficulties experienced with near vision by increasing age, known as presbyopia, is primarily attributed to the reduction in accommodation due to the rising stiffness of lens substance [9,10]. A potential treatment of presbyopia is the replacement of lens substance by functional artificial lenses, such as mechanically accommodating intraocular lenses or lenses filled with polymers [11,12,13]. These techniques, however, rely on the retained biomechanical function of the capsular bag, which is compromised during cataract removal and lens implantation surgery, to restore some or all of the accommodative process. When the lens substance is removed from the capsular bag similar to the surgery, it shows minor difference of the reaction force with the lens containing substance, addressing the primary biomechanical function of lens capsule [14].

The accommodative function of the crystalline lens has been studied with ex vivo testing and numerical modelling [15]. The shape change of lenses during accommodation in vivo can be replicated by inducing radial stretch using a lens stretching device [16,17,18]. The deformation and reaction force of lenses by such devices have been used to investigate the aetiology of presbyopia by combining optical power and biometry measurements [19,20]. Further, the biomechanical behaviour of lenses can be better understood by constructing finite element models [10,15,21,22,23,24]. Although the mechanical behaviour of the lens capsule itself has been studied by performing uniaxial or inflation tests [1,6], few studies directly investigate the behaviour of the capsule under accommodative forces. For the development of ophthalmic products treating presbyopia, it is essential to test any prototype inside the capsule to characterise how it will perform under different conditions; however, the access to human tissues is limited and animal capsules have different biomechanical properties compared to that of human capsules [6,14].

Thus, this study aimed to generate an alternative product able to mimic the biomechanical performance of human capsules, where a silicone rubber can be used to replicate the hyper-elastic response of human capsule [25]. The base material was characterised experimentally by mechanical tests at different conditions. A constitutive model was subsequently calibrated using the testing data and incorporated into a finite element analysis (FEA) model to simulate an accommodating test of an artificial capsule made from the same material. The characterisation together with the FEA modelling provided an insight into the intrinsic behaviour of materials and the mechanism of accommodation.

## 2. Materials and Methods

### 2.1. Material Preparation

An industrial-grade, room temperature vulcanised (RTV) silicone (Model: ZA13, Zhermack, Rovigo, Italy) was selected to be the base material. The silicone rubber had a low Shore A Hardness (*S*) (of 13) after curing with an equivalent Young’s modulus (*E*) of 0.5 MPa by Equation (1) [26,27], which was close to the lowest boundary of stiffness of human lens capsules ex vivo [1,3]. The material had a base fluid A and catalyst fluid B, which was mixed thoroughly at a ratio of 1:1. The catalyst fluid activated the condensation of the base fluid, allowing for a working time of 40 to 50 min at 23 °C. According to the operating instructions from the supplier, a vacuum was applied to the mixed fluid for 3 min to eliminate any air pockets. The degassed fluid was carefully poured into a mould and set for 24 h at room temperature (of 23 °C) to form a solid. Two types of strips were prepared for the mechanical characterisation, with dimensions of 6 mm × 25 mm × 10 mm (strip A) and 100 mm × 25 mm × 2 mm (strip B), respectively (Figure 1a). The moulded strips had stable shapes (Figure 1b) with a tolerance thickness of ± 0.08 mm.
(1)$$E=\frac{0.0981\left(56+7.66S\right)}{0.137505\left(254-2.54S\right)}\;,$$

### 2.2. Mechanical Characterisation

A universal mechanical testing machine (Model: 5942, Instron, Norwood, MA, USA) was used for material characterisation, which had a maximum loading capacity of 500 N, maximum extension of 488 mm, and speed between 0.05 and 2500 mm/min. The strip A was installed onto the machine for the uniaxial tension test, which was fixed at the bottom and elongated on the top by two respective grips (Figure 2a). An initial separation of 75 mm was assigned between the two grips as the gauge length. The testing protocols were defined by an integrated software Bullhill (Version: 3.0, Instron, Norwood, MA, USA). A monotonic tension (MT) was defined by applying a displacement of 90 mm at 100 mm/min, resulting in a nominal strain of 1.2 at 0.02 s^−1^. After the first primary loading (PL), a sagging state (of strip A) was observed when the sample returned to the initial separation with zero displacement (Figure 2b). This implied a negative force of the specimen under displacement-controlled deformation, recognised as the property change of rubber materials at repeated loading (Mullins effect). The process was repeated by performing reloading (RL) on the same specimen five more times.

After the monotonic test, a cyclic tension (CT) testing was performed on a new specimen by six cycles of loading and reloading operation at 100 mm/min. Four amplitudes of displacement (15, 37.5, 75, and 90 mm) were defined during loading, aiming for a final nominal strain of 0.2, 0.5, 1.0 and 1.2, respectively (at 0.02 s^−1^). A force-controlled unloading was defined to avoid sagging by halting the top grip when the reaction force was zero. In addition to the cyclic test at low speed (of 100 mm/min), another cyclic test was performed on a new specimen by increasing the speed to 1000 mm/min, which provided a condition with 10 times the strain rate (of 0.2 s^−1^). The repeatability of the material behaviour was checked by comparing the results at the different testing conditions.

Strip B was used for a simple shear test on the testing machine by designing customised fixtures (Figure 3a). Two long aluminium base bars were fixed by two clamps on the machine, offset by a gap of 6 mm, to allow the specimen to be installed along the elongation axis. It was bonded onto the surfaces of the bases by adhesive made of Ethyl 2-cyanoacrylate (Brand: Loctite control, Henkel, Winsford, UK). This application introduced a single lapped in-plane shear test (Figure 3b). By using the software (Bullhill, Version: 3.0), a monotonic shear (MS) test was defined by assigning a displacement of 8.4 mm at 8 mm/min (with gauge length of 6 mm), creating a maximum shear strain of 1.4 at 0.02 s^−1^. Similar to the tension test, one primary loading (PL) and five reloading (RL) operations were applied on the same specimen for MS. The cyclic shear (CS) test by force-controlled protocol was conducted on new specimens by applying four amplitudes of displacement (1.2, 3.0, 6.0, and 8.4 mm) at 8 mm/min to achieve different levels of maximum shear strain (of 0.2, 0.5, 1.0 and 1.4).

### 2.3. Accommodating Test

An artificial lens capsule (ALC) was manufactured by using the same batch of silicone material and moulding process (Figure 4a) [28]. The ALC had an anterior and posterior half with an average thickness of 150 ± 40 µm measured by calliper. It had an ellipsoidal shape with a radius of 4.5 mm, 1.6 mm, and 2.55 mm along the equator, anterior sagitta and posterior sagitta, respectively. An extension ring with a width of 0.5 mm and thickness of 1 mm was fitted around the equator to allow the ALC to be fixed onto a support structure, which has eight branches equally distributed (45° angle) and mounted on a lens radial stretching system (LRSS) (Figure 4b) [28]. Each branch had independent radial motion by performing a radial cut of the joint region. The LRSS provided stretch and release of each branch simultaneously radially to mimic the action of the ciliary muscles and adjoining zonules on the capsule. The equatorial stretch of the ALC was calibrated at two linear nominal speeds (NS) of 0.5 mm/s and 0.05 mm/s, driven by a stepper motor. A duration of 2.7 s and 27 s was needed, respectively, to offer a diameter change of 0.9 mm, which was similar to the displacement observed in vivo to a 10 D accommodation stimulus [29,30]. The ALC was filled with ophthalmic viscosurgical devices (OVDs) after it was mounted to the LRSS, which helped to recover and maintain the initial shape of the ALC under pressurisation. Preliminary cyclic testing was performed on the ALC at a high speed (NS = 0.5 mm/s) and then on the same ALC at a low speed (NS = 0.05 mm/s). The shape change of the ALC was monitored from the side view by a digital microscope camera (Model: Cmex18pro, Euromex, Arnhem, The Netherland) with a resolution of 20-million pixels. A load cell was installed on one of the arms of the LRSS to measure the reaction force of 3 tests at each condition. The stretching force was derived by extracting the nonlinear force contributed other than the ALC.

### 2.4. Constitutive Modelling

An isotropic polynomial incompressible strain function was used to model the constitutive behaviour of material (Equation (2)). The initial shear modulus (μ) can be estimated (Equation (3)) and the equivalent Young’s modulus (E) can be deduced based on the incompressibility of materials (Equation (4)).
(2)$$U=\sum_{i+j=1}^{N}C_{ij}{\left({\bar{I}}_{1}-3\right)}^{i}{\left({\bar{I}}_{2}-3\right)}^{j}+\sum_{i=1}^{N}\frac{1}{D_{i}}{\left(J-1\right)}^{2i}\;,$$
(3)$$\mu =2\left(C_{10}+C_{01}\right)\;,$$
(4)E≈3μ ,
where J is the determinant of the deformation gradient (F) denoting the volume change (Equation (5)), ${\bar{I}}_{1}$, ${\bar{I}}_{2}$ is the deviatoric first and second invariant of the right Cauchy-Green tensor (C) under finite deformation (Equations (6)–(8)).
(5)$$J=\det\left(F\right)\;,$$
(6)$$C=F^{T}F\;,$$
(7)$${\bar{I}}_{1}=J^{-1/3}\mathrm{tr}\left(C\right)\;,$$
(8)$${\bar{I}}_{2}=\frac{1}{2}J^{-1/3}\left[{\left(\mathrm{tr}\left(C\right)\right)}^{2}-\mathrm{tr}\left(C^{2}\right)\right]\;,$$

Three parameters ($C_{10}$, $C_{01}$, $C_{20}$) were calibrated in the constitutive model by tension and shear test data with the assumption of incompressibility (J ≡ 1). The calibration was based on a theoretical modelling of the deformation [31,32,33]. A least square approach (Equation (9)) was used to acquire the best fit of the stress-strain relationship by minimising the stress difference ($R^{2}$) between test data ($\sigma_{i}^{t}$) and modelling result ($\sigma_{i}^{m}$) under the same strain. The data from PL were used to fit the model across the whole strain level. To incorporate the Mullins effect, the cyclic RL data were used to fit another group of material parameters. As a small strain (of less than 0.14) was expected during accommodation [34], the RL data within the strain of 0.2 was employed and the residual strain was shifted to zero during calibration.
(9)$$R^{2}=\overset{N}{\underset{i=1}{\sum }}{\left(\sigma_{i}^{t}-\sigma_{i}^{m}\right)}^{2}\;,$$

A one-fourth geometric model of the artificial capsule surface was constructed for FEA by 3-node shell elements (S3) with an open-source meshing software (Salome, Version: 8.3.0, CEA & EDF, Paris, France) (Figure 5a). To represent the radial cutting of the joints, two groups of repeated nodes sharing identical coordinates were built, belonging to different elements of each side, which allowed the separation of each side under circumferential tension. Symmetric boundary conditions were defined at the edges of the plane along X = 0 and Z = 0. Loads were applied by controlling the relative amplitudes to replicate the accommodating test (Figure 5b). Within a time-amplitude of 0.1, a linear pressure load was applied on the interior surface of the capsule to simulate the application of OVDs. A trial-and-error of pressure value was conducted to yield a similar profile of the ALC to that measured in the experiment. A linear displacement from time-amplitude of 0.1 to 1 was provided along the outer equatorial edge to simulate the radial stretch. A static analysis was performed by an open-source FEA solver (CalculiX, Version: 2.17, Friedrichshafen, Germany) [35]. The total reaction force of the nodes between two radials cutting positions (of 45° angle) was exported and the stretching force was derived by the relative values between the time amplitude of 0.1 and 1.

## 3. Results

### 3.1. Monotonic Test

The mechanical behaviour of the silicone material was displayed by the stress-strain relationship under MT and MS test at 0.02 s^−1^ (Figure 6). At primary loading (MT_PL) (Figure 6a), the virgin material showed an evident hyper-elastic behaviour by a linear curve at low strain regime (< 0.6) and nonlinear curve beyond. Among the reloading process (MT_RL), there was an offset of stress-strain path with that of MT_PL, which is known as the Mullins effect of rubbers, implying the stress softening due to the internal damage of materials [36]. This corresponded to the sagging of the specimen where the initial motion of grip was to overcome the sagging with zero reaction force. Under MS at 0.02 s^−1^ (Figure 6b), the nonlinearity of stress-strain relationship was more evident during RL than that of PL. The offset between PL and RL indicated the existence of Mullins effect, whilst the residual strain was observed to be negligible.

### 3.2. Cyclic Test

The Mullins effect was further displayed by the result of cyclic tension (CT) and shear (CS) tests, with variable softening behaviour at different strain amplitude (Figure 7). It was found that the stress-strain relationship of MT_PL and MS_PL was along the response of the first primary loading (PL) path of each amplitude for both CT (Figure 7a) and CS tests (Figure 7b), by minor deviations. By increasing the strain amplitude, the stress softening behaviour became more evident, with elevated residual strain at zero stress. At high strain amplitude (of over 1.0), there was a slightly different mechanical response of materials along the reloading and unloading paths. The influence of strain rate was revealed by comparing the behaviour of material at two strain rates (of 0.02 and 0.2 s^−1^) under CT (Figure 7c). There was no marked difference on the stress-strain relationship along the PL and RL path by similar overall hyper-elasticity and stress softening behaviour, which indicated the irrelevance of mechanical behaviour to strain rate.

### 3.3. Modelling and Simulation

The material parameters of the constitutive model ($C_{10}$, $C_{01}$ and $C_{20}$ in Equation (2)) were calibrated by the data from monotonic and cyclic test along primary-loading (PL) and reloading (RL) paths (Table 1). The shear moduli (μ) and equivalent Young’s moduli (E) were calculated to obtain the initial stiffness of materials (Equations (3) and (4)). It was observed that the material at RL exhibited higher stiffness than that of PL; this was due to Mullins effect and more accurate fitting of the data points at small strain regime (of less than 0.2). The Young’s modulus at PL and RL was around 0.45 and 0.49 MPa, respectively, which was in accordance with the value (of 0.5 MPa) estimated by the hardness of material (of Shore A Hardness 13).

The performance of the constitutive model was displayed by comparing it to the experimental test (Figure 8). Along the primary loading path of monotonic uniaxial tension (MT_PL) and simple shear (MS_PL) at 0.02 s^−1^ (Figure 8a), there was a good consistency between experiment and modelling over the whole strain regime; this indicated the uniqueness of material parameters and their applicability for different modes of deformation. By using the reloading data under cyclic testing, the modelling results with two groups of material parameters (PL and RL) were compared with the experimental test (Figure 8b). Within an initial small strain (of 0.02), there was no evident diversity between the two models with PL and RL. As the strain increased, a slightly steeper stress-strain relationship was identified by the modelling with RL, which captured the material behaviour better. The contrast indicated the suitability of using modelling with PL only for virgin material, but with RL for the material after the initial stretch.

In FEA, a large bulk modulus (K = 150 MPa) was employed to satisfy the incompressibility, which was 1000 times the shear modulus (of 0.15 MPa), resulting in an equivalent Poisson’s ratio (of 0.4995) close to 0.5. An internal pressure of 2.0 kPa was found by trial and error to achieve a similar profile of the ALC pressurised with OVDs. The shape changes of ALC by FEA_PL and FEA_RL were illustrated by comparing the initial profile (mesh) and final profile (surface) (Figure 9a,b). By applying radial stretch, an increase of equatorial diameter was found, resulting in a decrease of sagittal distance at anterior and posterior half. A homogeneous strain area covering a big region of the anterior and posterior surface, indicated an approximate strain magnitude of 0.15 (of anterior half) and 0.065 (of posterior half). There was a marked stress-intensive region near the cutting line of the joints, with a strain magnitude of 0.2. There was no evident difference in the deformation between FEA_PL and FEA_RL, which was primarily due to the incompressibility of the material under displacement-controlled deformation.

As there was similar shape change by FEA, the historic profile of the ALC by FEA_RL was compared to the experiment tests (Figure 9c,d). During the preliminary test of the ALC at high speed (NS = 0.5 mm/s) (Figure 9c), the initial shape of the ALC was very well represented by FEA, implying the pressure load applied to the modelling was suitable to simulate the application of OVDs. Comparatively, there was less inflating state of the ALC with secondary testing at low speed (NS = 0.05 mm/s) than with FEA (Figure 9d), which could be attributed to the pressure loss after the preliminary cyclic tests. Under the displacement history of the LRSS, there was a minor change of profile at the low displacement (of 0.1 mm) at the initial stretch (t < 1.5 s at NS = 0.5 mm/s and t < 15 s at NS = 0.05 mm/s) between the experiment and FEA. The subsequent stretching process from the experiment was well predicted by FEA under equivalent stretch, implying similar anterior and posterior curvatures and sagittal distance.

The shape change of the ALC was further compared by fitting the initial and final profile with circular arcs to acquire the curvature radius of anterior (R_A_) and posterior (R_P_) surfaces and sagittal distance (Figure 10a,b). In the FEA modelling, the curvature radius before stretch was found to be R_A_ = 6.9 mm and R_P_ = −5.3 mm with a sagittal distance of 4.9 mm. Compared to the experimental test at NS = 0.5 mm/s (Figure 10a), a curvature radius of R_A_ = 7.3 mm and R_P_ = −5.4 mm with a sagittal distance of 4.7 mm was observed, implying a good correspondence of curvatures and minor over-inflation at the posterior half (Figure 10a). The pressure loss of the ALC during experimental testing at NS = 0.05 mm/s was exhibited by the initially lower inflation compared to FEA, revealing a result of R_A_ = 7.0 mm and R_P_ = −4.8 mm with a sagittal distance of 4.6 mm (Figure 10b). After stretch, FEA predicted a profile of R_A_ = 10.7 mm (ΔR_A_ = 3.8 mm) and R_P_ = −6.9 mm (ΔR_P_ = 1.6 mm), with a sagittal distance of 3.9 mm (Δ = 1 mm). The experimental result after stretch at NS = 0.5 mm/s was ΔR_A_ = 2.1 mm, ΔR_P_ = 1.0 mm and decrease of sagittal distance of 0.6 mm, indicating a smaller change than with FEA modelling (Figure 10a). At the secondary test conducted at NS = 0.05 mm/s (Figure 10b), the stretch introduced a profile change of ΔR_A_ = 2.3 mm, ΔR_P_ = 0.9 mm and a decrease of sagittal distance of 0.7 mm, resulting in a profile closer to the FEA modelling.

Despite the similar change of profile of the ALC by FEA_PL and FEA_RL, there were different reaction forces from FEA simulation. At high speed (NS = 0.5 mm/s) (Figure 10c), both FEA modelling with PL and RL indicated a similar linear reaction force versus the displacement to the testing data from three repeats. Within a displacement of 0.2 mm, the deviations between FEA_PL and FEA_RL were very small. At higher stretch, there was more correspondence with the testing data by FEA_RL, implying better applicability of the material parameters. In the secondary test conducted at NS = 0.05 mm/s (Figure 10d), a small deviation was observed between test and modelling, within a small displacement of 0.1 mm, due to a higher reaction force from the experimental test. The steeper curve indicated a stronger force than that at a high speed and FEA modelling, regardless of the rate independence of the material. This was probably attributable to the status of the ALC with pressure loss, introducing bigger turbulence force from the shape change under radial stretch. As the displacement reached 0.2, a decaying reaction force was observed during the testing process, indicating the existence of relaxation that was not incorporated in FEA, leading to an apparently higher reaction force than in the experimental test.

## 4. Discussion

Under small strain regime (of less than 10%), the mechanical response of the human capsule is usually assumed to be linear-elastic with a Young’s modulus of 0.7 to 1.5 MPa [37,38]. This assumption conformed to the results (of 0.4 to 1.5 MPa) under uniaxial tension of the human’s capsule ex vivo [1,3,39]. The silicone rubber for the artificial capsule was selected with a stiffness equivalent to a modulus (of 0.5 MPa), close to the lowest limit of human capsule under uniaxial tension. Different values beyond this range have been reported with higher values (of 2.0 to 8.0 MPa) and lower values (of 0.03 to 0.3 MPa) [2,40]. Although the liner-elastic model provided similar results to the hyper-elastic model for the inflation test of capsule [37,38], there was a large disparity with the modelling of uniaxial tension [41]. This highlighted the need for characterising a material by using different modes of deformation. The mechanical nonlinearity of the human capsule turned more significant with a high strain regime [1,3,39], where hyper-elastic models became more appropriate [15,24,42]. More complex behaviours of the biological capsules, such as stress softening, viscoelasticity (creep, relaxation) and anisotropy, exist at different loading conditions [1,43,44,45,46,47,48,49]. The stress softening, i.e., Mullins effect, was observed for silicone rubber under cyclic loading, and was more evident at higher strain levels [36,50]. A complete understanding of this behaviour relies on more comprehensive testing and modelling [51,52], not conducted in the current study but was simplified by using two groups of material parameters to take account of the reloading data. The viscoelasticity of silicone rubber, such as the rate dependence, was reported to be negligible [53,54], which was proved by the uniaxial tension test at different speeds.

The ALC in free state had an equatorial radius (of 4.5 mm) close to the biometry of the human capsule at accommodated state (of 4.5 to 5.0 mm) [10,14,19,55,56]. There was a definite sagittal distance of the FEA model (of 4.9 mm) similar to an aged biological capsule (of 4.8 mm at 45 years old) [14,19,55]. These dimensions have been widely employed for FEA modelling of accommodation [37,38,57]. A big difference of spatial thickness existed between the ALC and the human capsule. A homogenous membrane with an average thickness (of 150 µm) was employed in the FEA model, whilst the human capsule has a much smaller thickness (of 5 to 30 µm) and large inhomogeneity [1,8,25]. This relatively large thickness of the ALC was beneficial for the moulding operation and prevention of collapse [28]. In the accommodating simulation, Helmholtz’s theory was observed to change the shape of capsule of the zonules, which was driven by the ciliary muscle [58]. The zonules have been found to locate on the anterior, equatorial, and posterior region of the capsule by inhomogeneous distribution and implied the dependence on age [59,60]. These properties can be defined with different arrangement and stiffness in the modelling [15,34,37,38,61,62], where the influence was found to be significant for displacement-controlled condition [37,61], but negligible for force-controlled condition [38]. Compared to the zonule distribution of a biological model, the ALC model was simplified significantly by placing the attachments around the equatorial region with a similar width to that found in the human lens (of 0.2 to 0.4 mm) [15,37,38]. The stiffness was strongly enhanced by using a relatively large thickness (of 1 mm) to transfer the radial stretch more effectively whilst the circumferential force was eliminated by performing radial cuts as suggested by other studies [17,18,63].

The accommodation of eyes can be modelled numerically by using the capsule only with the application of pressure from lens substance [7,24,42,62]. The relative pressure from the lens substance was defined to be 4 to 6 mmHg (of 0.5 to 0.8 kPa) under accommodative stimulation [24,42], whilst the resisting pressure of the lens capsule can reach as high as 40 mmHg (of 5.0 kPa) [48]. A high pressure value (of 2.0 kPa) of the ALC was deduced by a reverse approach with FEA to be compensated for its high thickness (of 150 µm) in contrast to the human capsule (of 5 to 30 µm), providing the insight into the over-inflation status in a previous study [28]. Compared to the studies using a flat surface as the initial shape of the human capsule, the reference state of the ALC was defined to be the moulded shape, significantly diminishing the pretension effect of materials before exerting stretch. The curvature radius of the pressurised ALC in the FEA model (R_A_ = 6.9 mm, R_P_ = −5.3 mm) was in accordance to the human capsule of young age (of 19 to 29 years) [10,21,22]. The modelling resulted in the surface of posterior capsule after stretch showing a curvature radius (R_P_ = −6.9 mm) consistent with the in vivo study and simulation [10,15,21,22,23,24]. The deformed surface of the anterior capsule (R_A_ = 10.7 mm) had a curvature radius between that reported for young (of 12.0 mm) and old (of 9.0 to 9.8 mm) lenses [19,64]. To achieve the change on curvature of radius aforementioned and sagittal distance (of 1 mm) for the ALC, a radial displacement (of 0.5 mm) was exerted onto the equator in the experimental test and FEA simulation. In contrast, a higher radial stretch (of 2 mm) was employed to the rigid part in the ex vivo stretching test of human lenses [14,17,20,55,65]. An evident accommodative power change has been observed by radial stretch (of 0.8 to 1.0 mm) on the ciliary body [66]. MRI studies have shown the adjustment of lens diameter is between 0.3 and 0.6 mm and lens thickness between 0.1 and 0.6 mm under binocular accommodative stimulus conditions (of 0.1 to 8.0 D) [19,21,29,64]. These findings have been replicated by the modelling of accommodation with FEA [15,37,38,57,62,67,68], the approach of which was employed in the presented FEA modelling and further advanced by the accompanied validation from experimental testing.

The contraction force was a resulting parameter for the displacement-controlled deformation driven by the radial stretch [37,57,62,68], which can also be explicitly defined as the stimulus for load-controlled deformation [15,24,38,67]. Experimental and numerical studies have shown a single or multiple linear relationship between the stretching force and radial stretch [14,17,18,19,37,68]. The multilinear correlation was attributed to the influence of testing environmental factors (mobilisation, pretension, and dynamic effects) during the stretching process by a previous study [28], which was further validated by the linearity and accordance of FEA by extracting the force contribution only from the stretch of the ALC. The overall contraction force of the human lens ranged from 3 g to 13 g as assessed by ex vivo stretching tests [14,17,18,19,69], which is slightly reduced by the extraction of lens substance [14,18]. Most numerical studies showed a similar stretching force over a wide range (of 2 to 14 g) [15,24,37,38,57,62,67,68]. A larger stretching force (of 20 to 30 g) required for the ALC from experiment and FEA, was primarily due to the higher thickness of ALC (of 150 µm) compared to the human lens capsule (of 5 to 30 µm) [1,2,6], which needs to be further reduced in order to provide a smaller average overall stretching force.

## 5. Conclusions

An artificial lens capsule (ALC) based on soft silicone has been developed to replicate the mechanical behaviour of lens capsule during the accommodating process. A hyper-elastic behaviour of the base material together with stress softening (Mullins effect) and rate independence were demonstrated by mechanical characterisation under different modes of deformation. The constitutive model was calibrated using experiments, which captured the mechanical behaviour by numerical modelling. The accommodating test of the ALC was successfully modelled by using FEA incorporating the constitutive model. The deformation of the ALC showed convincing similarity to the human capsule by both experimental testing and modelling. The applicability of the FEA modelling was highlighted by presenting the correspondence of the deformation; this can be used to adjust the thickness to correct for the discrepancy of the contraction force of the ALC due to its manufactured thickness profile. The development of the ALC with relevant mechanical behaviour will facilitate the examination of ophthalmic implants for overcoming presbyopia implanted within capsular bag, such as accommodating intraocular lenses (AIOLs). The FEA modelling provides a convenient approach to examine and validate the function of implants within the human capsular bag of variable stiffness and geometry.

## Figures and Tables

**Figure 1 polymers-13-03916-f001:**
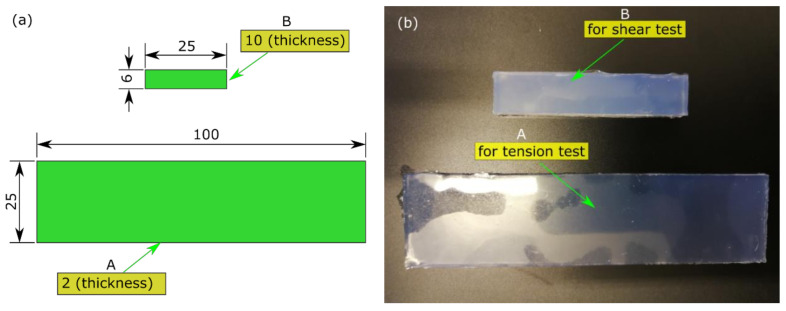
Testing specimens for characterisation: (**a**) dimensions (unit: mm); (**b**) moulded strips.

**Figure 2 polymers-13-03916-f002:**
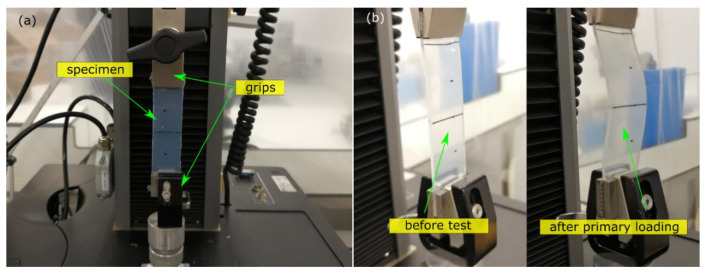
Uniaxial tension test: (**a**) testing setup; (**b**) shapes of specimen before test and after primary loading.

**Figure 3 polymers-13-03916-f003:**
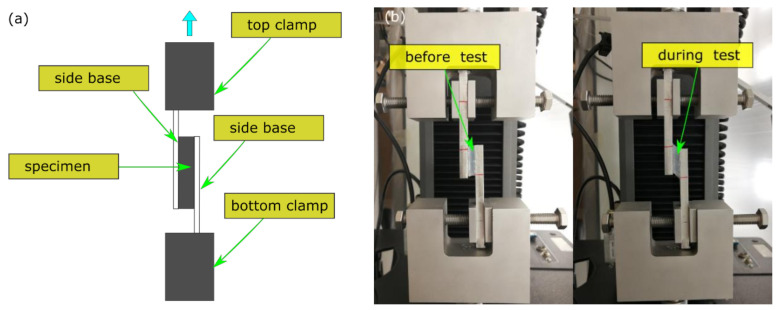
Simple shear test: (**a**) schematic testing setup; (**b**) shapes of specimen before and during test.

**Figure 4 polymers-13-03916-f004:**
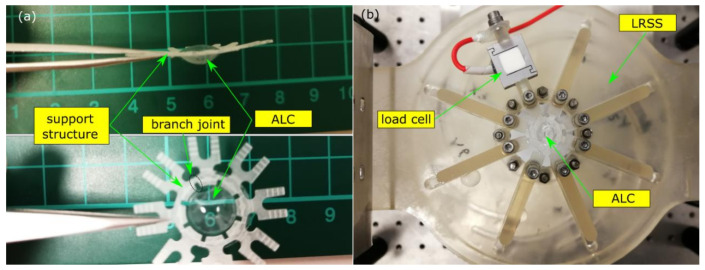
Accommodating test: (**a**) an artificial lens capsule (ALC) with support structure; (**b**) ALC on the lens radial stretching system (LRSS).

**Figure 5 polymers-13-03916-f005:**
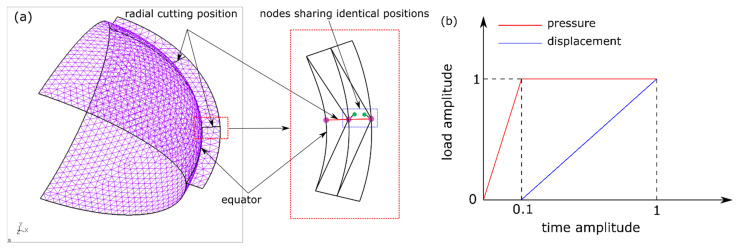
Modelling on accommodating test of ALC: (**a**) FE model; (**b**) loading step.

**Figure 6 polymers-13-03916-f006:**
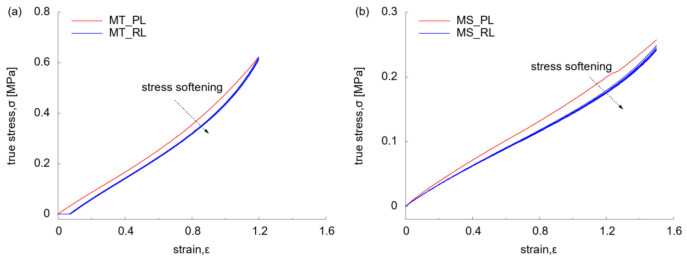
Stress-strain relationship under monotonic test: (**a**) at uniaxial tension (0.02 s^−1^); (**b**) at simple shear (0.02 s^−1^).

**Figure 7 polymers-13-03916-f007:**
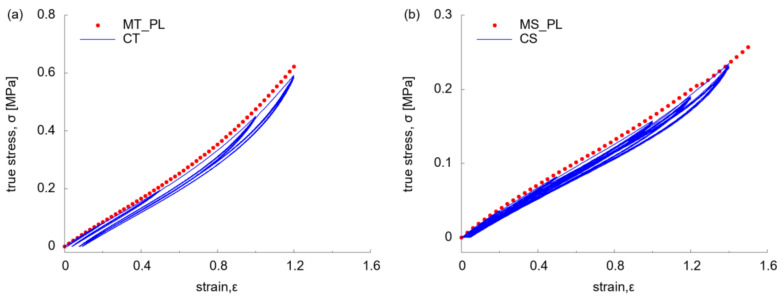
Stress-strain relationship under cyclic test: (**a**) at uniaxial tension (0.02 s^−1^); (**b**) at simple shear (0.02 s^−1^); (**c**) at uniaxial tension (0.02 s^−1^ vs. 0.2 ^s−1^).

**Figure 8 polymers-13-03916-f008:**
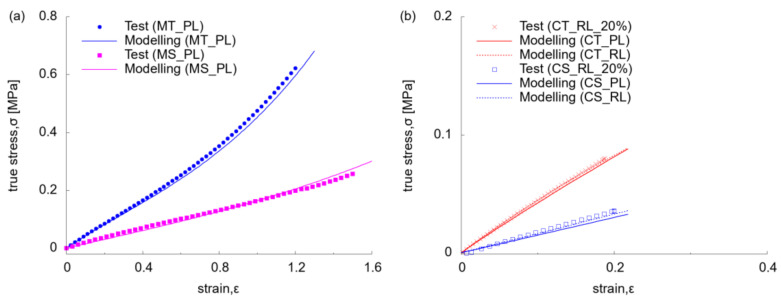
Comparison of stress-strain relationship between experiment and modelling: (**a**) primary loading under tension and shear test (at 0.02 s^−1^); (**b**) reloading under tension and shear (strain < 0.2 at 0.02 s^−1^).

**Figure 9 polymers-13-03916-f009:**
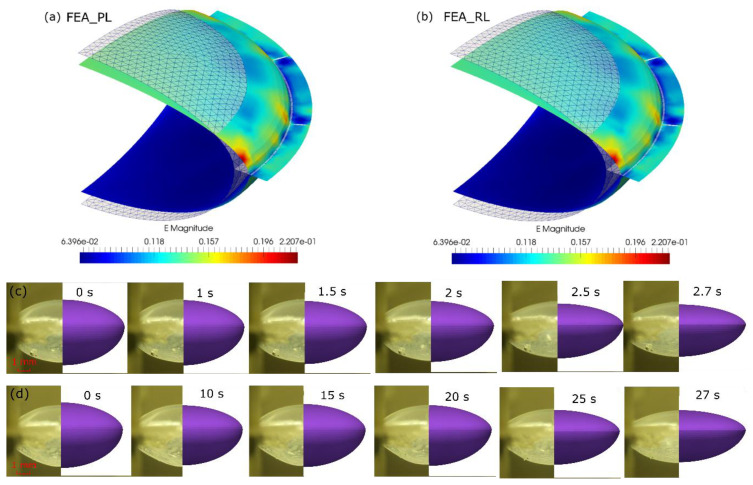
Shape change of ALC under accommodating: (**a**) initial shape and strain profile of deformed shape by FEA _PL; (**b**) initial shape and strain profile of deformed shape by FEA_RL; (**c**) comparison between experiment and FEA_RL (NS = 0.5 mm/s); (**d**) comparison between experiment and FEA_RL (NS = 0.05 mm/s).

**Figure 10 polymers-13-03916-f010:**
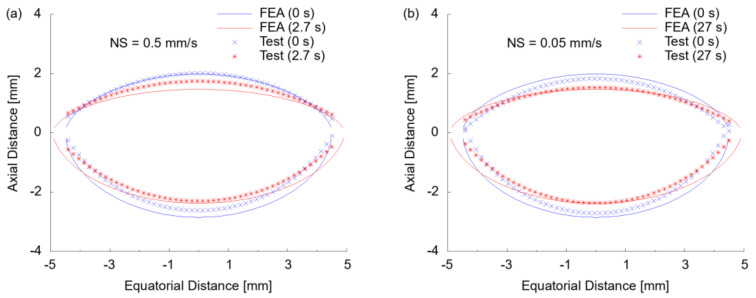
Comparison of profile and reaction force of ALC between experiment and FEA modelling: (**a**) change of profile at high speed (NS = 0.5 mm/s); (**b**) change of profile at low speed (NS = 0.05 mm/s); (**c**) radial stretching force at high speed (NS = 0.5 mm/s); (**d**) radial stretching force at low speed (NS = 0.05 mm/s).

**Table 1 polymers-13-03916-t001:** Material parameters of the constitutive model (unit: MPa).

	$C_{10}$	$C_{01}$	$C_{20}$	μ	E
PL	2.426 × 10^−2^	5.015 × 10^−2^	3.791 × 10^−3^	1.488 × 10^−1^	4.464 × 10^−1^
RL	5.000 × 10^−8^	8.160 × 10^−2^	5.000 × 10^−6^	1.632 × 10^−1^	4.896 × 10^−1^

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
