# Peer review of "Characterisation and Modelling of an Artificial Lens Capsule Mimicking Accommodation of Human Eyes"

_polymers, 2021, doi:10.3390/polym13223916_

Round 1

Reviewer 1 Report

The manuscript entitled "Characterisation and modelling of an artificial lens capsule mimicking accommodation of human eyes" is a significant study that provides useful information for research in health field. The manuscript is well written and the results are clearly presented. The methodology is robust, and the experiments were carefully done.

A few minor recommendations are listed below:

  1. The abstract section should be re-organized. It seems too generic, and it does not adequately summarize the results of the study.
  2. Current options in this field present limitations that lead to the need to expand and development of knowledge and, in this sense, from my point of view, the limitations of this study should be discussed, so as to open new directions. Additionally, the approach enables relatively control over material’s features.

Reviewer 2 Report

1. Introduction

In the line 49 "combing" should be changed to "combining".

2. Materials and methods

What was the catalyst used for?

Was the material purified after reaction with catalyst?

Was the catalyst left in the material analyzed?

Was the material analyzed to confirm the appropriate chemical structure?

Was the chemical structure of prepared strips and lens compared?

Round 2

Reviewer 2 Report

The author response is sufficient.